# OpenReview forum: "LM Agents May Fail to Act on Their Own Risk Knowledge"
_ICLR.cc/2025/Workshop/BuildingTrust — BuildingTrust_

### Official Review · Reviewer_swpq · 2025-02-25
**Summary**

**Rating:** 7
**Confidence:** 3

**Review:**

### Summary of the Paper:
The authors of this paper evaluated the safety behaviour of AI agents across three self-defined and created frameworks. These frameworks assess the agents' basic knowledge about safety, their ability to identify risks in working trajectories, and their final execution in dangerous situations. Moreover, they created a risk verification framework for analyzing potentially risky actions.

### Strengths:
Dividing the evaluation into three different benchmarks is reasonable. The paper clearly presents the evaluation methods and describes the assessment process well.
Moreover, the authors proposed a solution for mitigating the gaps between knowledge-identification and identification-execution stages.

### Weaknesses:
The initial 144 test cases and the final 328 trajectories might not adequately cover all potential dangerous situations and behaviors across AI agents.

### Questions and Suggestions for Improvement:
- I didn’t notice which model you used for the safety and helpfulness evaluator. Did you provide any manual review of the obtained scores?
- Similarly, I didn’t see which model you used as the LLM-based risk context extractor (Section 3.2).
- Did you test the system on benign trajectories that share many similarities with unsafe ones? Do AI agents based on LLM models behave properly in these test cases?

---

### Official Review · Reviewer_uWmd · 2025-03-01
**Interesting perspective on safety**

**Rating:** 6
**Confidence:** 1

**Review:**

Strengths:
- This paper seems to present a novel setup focusing on software/terminal executions for agents.
- It is also interesting to distinguish between the identification and the execution ability.

Weaknesses:
- I find the paper pretty difficult to understand, but must also clarify that I am an uninformed outsider.

---

### Official Review · Reviewer_skKj · 2025-03-01
**The work is both timely and valuable by identifying a crucial knowledge-action risk gap in emerging agentic systems and proposing a baseline defense to mitigate this gap.**

**Rating:** 6
**Confidence:** 4

**Review:**

Pros:
1. By extending prior generator-validator concepts into scenarios involving malicious requests, this paper offers important insights by systematically categorizing LM agents’ failures into three progressive levels. This framework could potentially serve as an evaluation suite for future safer agentic system designs.

2. It concludes with actionable guidance—such as incorporating ToolEmu tool-using trajectories—to enhance the safety alignment of LLMs.

3. The presentation and writing style is straightforward and easy to follow.

4. While the work builds heavily upon the ToolEmu paper, it provides valuable insights by broadening the scope to more realistic settings. It highlights how agents can be aware of a harmful action yet fail to identify it within a trajectory, which can potentially translate into malicious execution.

Cons:
1. More details about the effectiveness of the abstractor are needed. For the proposed knowledge test, trajectories are transformed into objective descriptions for QA evaluation. However, the degree of transformation accuracy for the abstractor are not clearly presented in the paper.

2. Most experiments rely heavily on the ToolEmu environment; incorporating additional frameworks and more diverse, real-world datasets could improve the generalizability of the findings.

3. The proposed defenses (1) safety prompts and (2) LLM-based critique, while effective in standalone LLM settings, are relatively straightforward and might warrant further innovation for broader or domain-specific applications.

Overall, I would suggest accepting the paper since it identifies important findings on the knowledge-action gap in agents and proposes a simple yet effective baseline defense for mitigation.

---

### Decision · Program_Chairs · 2025-03-04

**Decision:**

Accept

**Comment:**

The paper heavily relies on the ToolEmu environment, limiting the generalizability of its findings, and lacks diverse, real-world datasets to validate its proposed framework. Additionally, the effectiveness of the abstractor in transforming trajectories for QA evaluation is not clearly demonstrated, and the proposed safety defenses, while useful, are relatively straightforward and could benefit from further innovation.